# Comparison of Diagnostic Profiles of Deaf and Hearing Children with a Diagnosis of Autism

**DOI:** 10.3390/ijerph20032143

**Published:** 2023-01-24

**Authors:** Rachel Hodkinson, Helen Phillips, Victoria Allgar, Alys Young, Ann Le Couteur, Andrew Holwell, Catarina Teige, Barry Wright

**Affiliations:** 1Leeds and York Partnership NHS Foundation Trust, Leeds LS7 3JX, UK; 2Peninsula Medical School, University of Plymouth, Plymouth PL4 8AA, UK; 3School of Nursing Midwifery and Social Work, University of Manchester, Manchester M13 9PL, UK; 4Neurodevelopment and Disability Group, Newcastle University, Newcastle upon Tyne NE1 7RU, UK; 5South West London and St. Georges Mental Health Trust, London SW17 0YF, UK

**Keywords:** autism, autistic, autistic spectrum disorder, autistic spectrum condition, deaf, child, young person, semi-structured interview

## Abstract

There is limited research comparing the presentation of autism in deaf and hearing children and young people. These comparisons are important to facilitate accurate diagnosis, as rates of misdiagnosis and delay in diagnosis amongst deaf children and young people are high. The aim of this study was to compare diagnostic assessment profiles of a UK cohort of autistic deaf and hearing children and young people. The Autism Diagnostic Interview—Revised—Deaf adaptation was completed with the parents of 106 children and young people (deaf children = 65; hearing children = 41). The majority of items explored showed no significant differences between deaf and hearing children and young people. Differences were found in peer relationships, where autistic deaf participants were less likely to respond to the approaches of other children or play imaginatively with peers. These findings need to be taken into consideration by clinicians in the assessment process.

## 1. Introduction

There are at least 53,954 deaf children in the UK; according to the Consortium for Research into Deaf Education (CRIDE) report of 2019, nine per cent of these children use British/Irish sign language [1]. Although precise epidemiological data are not available, previous research from the US suggests estimates of 1–2% of deaf children and young people have an autism spectrum disorder (ASD) [2,3]. Autism encompasses a broad range of characteristics across two main symptom behaviour domains required for a diagnosis [4]. Alongside this, there are individual profiles of skills and difficulties and relatively high rates of co-morbidities [5], including children who are deaf [1]. There is limited research exploring the presentation of autistic deaf children compared with autistic hearing children.

Comparisons of presentation in deaf and hearing autistic children are important to facilitate accurate diagnosis of autism in deaf children. This group is an at-risk population because of increased rates of syndromes and broader neurological problems related to being deaf [6,7].

Diagnostic errors including misdiagnosis and/or delay in diagnosis are also reported in this population [6,7,8,9,10,11,12,13].

The median age of autism diagnoses in hearing children in UK is 55 months [14] whilst the mean age in the US varying between studies, from 38 and 120 months [9,15]. Brett and colleagues [14] have suggested that some of the reasons for delayed diagnosis of autism in the UK include complex pathways to assessment and diagnosis, parental lack of awareness of autism and lack of awareness in some professionals. There are difficulties for deaf children and their families seeking diagnosis, as research has identified a number of barriers to accessing assessments [10] and considerable lack of confidence and expertise amongst professionals assessing deaf children for autism in the UK [11]. Barriers included the fact that existing assessments are heavily influenced by norms from hearing cultures, for example in the many different ways that metaphors are used in communication [16]. Other barriers included accessing services that could communicate with parents and children in order to facilitate a valid assessment, confusion by parents and professionals about whether symptoms were related to autism, being deaf or both and lack of understanding of the impact of autism on spoken or signed languages. Additionally, an estimated 90% of deaf children are born to hearing parents, who may not be expecting to have a deaf child [12,17], and so some parents may be uncertain what to look out for. Deaf children’s experiences of early communication can vary greatly [13], which, for some children, can lead to delays in exposure to and development of language as well as socio-emotional skills [18,19,20]. These potential risk factors may also be conflated by the emergence of early indications of autism [3]. The use of existing gold-standard assessment tools delivered in sign language fails many deaf children because they are often being assessed by a person communicating in another language working with an interpreter and there are numerous linguistic, cultural and sensory differences between deaf children and hearing children, including large differences in sensory processing, language and communication (e.g., visual prosody rather than verbal prosody) [7,13,21,22,23,24].

Previous studies have sought to identify some of the differing presentations of autistic deaf children [3,25,26,27]. However, few of these have been comparative in nature and most relied on measures developed with hearing children and young people in mind [28].

This study set out to contribute to this lack of information by comparing the diagnostic assessment symptom profiles of a UK cohort of autistic deaf and hearing children who had each received a clinical diagnosis of autism. The data were gathered using a validated standardised semi-structured interview undertaken with the main caregiver (parent) as part of a study adapting and validating this diagnostic measure for use in deaf children and young people [21]. Based on previous literature, we hypothesised that presentations of autism would be the same for both deaf and hearing children with some notable exceptions. We hypothesised that since autistic deaf children who use sign language sometimes reverse signs in American Sign Language (ASL) [26], this would be similar in British Sign Language (BSL). We also hypothesised that because of the integral nature of facial expressions to BSL, both to convey emotion but also because of its linguistic features related to prosody, use of facial expressions may differ between groups [25]. Given that a relatively high proportion of deaf children attend mainstream schools, we also thought we might see differences in peer relationships, with deaf children experiencing more challenges in this domain due to communication difficulties with their classmates. We sought to explore any differences in these areas.

## 2. Materials and Methods

### 2.1. Participants

Participants (parents of autistic deaf and hearing children with a clinical diagnosis of autism) were recruited by the UK Diagnostic Instruments for Autism in Deaf Children Study (DIADS) [29]—a national study to adapt and undertake initial validation of a set of standardised ASD screening and assessment measures including the Autism Diagnostic Interview—Revised (ADI-R) [30].

Participants were recruited through a variety of means. All schools for deaf children, mainstream schools with specialist resources for deaf children and special educational needs schools in England were contacted and asked to circulate study information to the parents or guardians of potentially eligible children. The ten specialist National Deaf Child and Adolescent Mental Health Services (NDCAMHS) centres across England [31] were also asked to circulate study information to potentially eligible families who had or were currently utilising their service. In addition to these services (and in order to recruit autistic hearing children as well as autistic deaf children), several organisations agreed to share information about the study with their members. This included the National Autistic Society, the National Deaf Children’s Society and the ASD-UK and DASLNE’s (Database of Children with Autism Spectrum Disorder Living in the North East) research databases. In addition, the study was advertised on relevant social media platforms such as the Child Oriented Mental Health Intervention Centre (COMIC) website, Limping Chicken (an online blog aimed at the Deaf community), as well as several online parenting groups.

A total of 78 autistic deaf children and 55 autistic hearing children were recruited, out of these 65 and 41 (deaf and hearing children, respectively) completed the ADI-R Deaf adaptation. Data from these participants were used to compare the profiles of deaf children and young people who had a NICE-guideline-compliant assessment giving a diagnosis of autism (see Section 2.2 and Section 2.3).

#### Inclusion and Exclusion Criteria

Participants were required to be aged between 2 and 18 years. For the purposes of the main study, deaf children/young people were required to possess a hearing range no higher than 40 dB HL. Both hearing and deaf participants were required to have evidence of an existing clinical diagnosis of autism (meeting the international diagnostic classification criteria for ICD-10/DSM-5, meaning a validated assessment including ADI-R, DISCO, 3di). They all underwent a NICE-guideline-compliant assessment including a validated assessment described in Section 2.2. No exclusions were made on the basis of health or mental health co-morbidities, intellectual disability, or language preference.

### 2.2. Instrument

The Autism Diagnostic Interview—Revised (ADI-R)—Deaf adaptation [21] is a 93-item semi-structured interview, adapted for use with deaf children and young people from the Autism Diagnostic Interview—Revised [24]. The interview was conducted by a trained interviewer with an informant, usually a parent/care giver, who knows the child/young person very well. Focussing on key domains of development and functioning (current concerns, early childhood development, acquisition and loss of language and other skills, communication and language functioning, social development and play, interests and behaviours and general behaviours), the interview allows clinicians to gather a detailed profile of the child/young person’s development by exploring early childhood and infancy, significant milestones, current behaviours and behaviours displayed when the individual was aged 4–5 years old, or in some cases exploring if they have ever displayed a given behaviour. Scores from specified interview items can then be combined into an instrument algorithm across three symptom domains: reciprocal social interaction, language and communication and repetitive and stereotyped patterns of behaviour. Exceeding the cut-off in all three domains indicates an ADI-R diagnosis (not the same as a clinical diagnosis) of autism [32] provided that developmental differences were noted before the age of three years.

### 2.3. Procedure/Analysis

Deaf children with and without a diagnosis of autism were recruited as described in a previous paper [21,22]. This followed strict inclusion criteria for autism and also for the comparison group including the instrument described in Section 2.2 for all participants. This allowed for analysis of sensitivity and specificity of the ADI-R Deaf adaptation; details of this analysis may be found in the adaptation and validation study’s main paper [21]. Parents/carers of participating children/young people were interviewed by clinicians trained in delivering the ADI-R Deaf adaptation [21] in a location convenient for parents/carers or over the phone where meeting in person was not feasible due to family and or clinician availability. Clinicians were matched to families based on language and communication preferences; for example, where deaf parents/carers used British Sign Language (BSL), a deaf clinician who also used BSL completed the interview with them; where this was not possible, qualified BSL interpreters attended.

To be able to meaningfully compare items and scores, the ADI-R Deaf adaptation was used for interviewing parents of both deaf and hearing children. Each item of the ADI-R Deaf adaptation is scored as defined in the original ADI-R [30], broadly reflecting: 0 = no difficulties according to the definition for that item, 1 = some difficulties and 2 = significant difficulties. Each item is further asked for two time points; this being ‘currently’ [defined as over the last 3 months] and ‘over the 12 month period between age 4–5 years’. For some items the instrument asks whether the behaviour was ‘most abnormal in the 12 months between 4 and 5 years old’. Some items ask this question as ‘ever’ occurring regardless of the participant’s age. The scores for the pre-specified items that contribute to the ADI-R algorithm framework were transposed to calculate each of the algorithm symptom domain scores. The algorithm item scores for the deaf and hearing children/young people were compared to identify any differences in symptom profile between the two groups.

Clinicians carrying out the interviews were also asked to comment on any other factors reported by parents or behaviours or symptoms noted by them as clinicians.

Summary statistics are presented as *n* (%) for each item. The term ASD is used throughout results tables to mirror the language of the respective items of the instrument used. Chi-square tests were used to compare between deaf and hearing children. A *p*-value of <0.05 was considered to indicate statistical significance. All analyses were performed on SPSS (IBM Corp. Released 2019. IBM SPSS Statistics for Windows, Version 26.0. Armonk, NY, USA: IBM Corp). No correction was used for the large number of comparisons.

## 3. Results

ADI-R Deaf adaptation interviews were completed with the parents/carers of 106 children/young people (autistic deaf *n* = 65, autistic hearing *n* = 41).

### 3.1. Demographics

The demographic information of participating children and young people can be seen in Table 1.

The ADI-R Deaf adaptation algorithm domain scores for the deaf and hearing children is presented in Table 2. Overall, the domain total scores for the deaf children were not statistically significant compared to hearing children.

Using the same cut-off scores as published by Rutter et al. [30] and as recommended in the initial validations study [21], the proportion of each group that meet the ADI-R diagnosis of autism is shown in Table 3.

### 3.2. Domain A: Reciprocal Social Interaction

#### 3.2.1. Use of Non-Verbal Behaviours to Regulate Social Interaction

There were no significant differences between deaf and hearing children and young people’s use of non-verbal behaviours to regulate social interaction (Table 4) for either current direct gaze [defined as over the last 3 months] (*p* = 0.507) and ‘most abnormal over the 12 month period between age 4–5 years’ (*p* = 0.640) or social smiling (current: *p* = 0.231, most abnormal: *p* = 0.546).

Similarly, no significant difference (*p* = 0.657) was noted in current use of facial expressions, with 17 (26%) deaf children using a full range of facial expressions compared to 10 (27%) of hearing children, nor when parents were asked to consider their children’s most abnormal behaviour when aged four to five years, (*p* = 0.539).

#### 3.2.2. Peer Relationships

Interesting differences were noted between the two groups when considering peer relationships.

Among the children in our study who were aged under 10 years at the time of the parent interview, significantly more autistic deaf children (*p* = 0.009) were reported to experience difficulties in playing imaginatively with peers compared to autistic hearing children (Table 5). In contrast, no such differences were noted by parents when they remembered back to their child’s play at age four to five years old (*p* = 0.177).

Significant differences were identified between the two groups when parents considered their child’s interest in other children when aged between four and five years of age (*p* = 0.030). Autistic deaf children and young people were over two times more likely to be reported as showing limited interest in peers (48 (77%)) than autistic hearing children in our sample (21 (53%)) (*p* = 0.03).

Similarly, autistic deaf children appeared significantly more likely at both time points (currently aged under 10, *p* = 0.005; at aged 4–5, *p* = 0.023), to be reported as rarely or never responding to the approaches of even familiar children or avoiding these approaches altogether (Table 6).

No significant differences were recorded in either children’s engagement in group play with peers, or in their ability to make friends.

#### 3.2.3. Shared Enjoyment

There are three items included in this section and no significant differences were noted between the groups when parents were asked to consider how their children showed and directed the attention of others, shared objects or shared enjoyment.

#### 3.2.4. Socioemotional Reciprocity

Significant differences were noted between the two groups when parents were asked to consider if their children had ever used another’s body to communicate (*p* = 0.012). Parents reported that over half of the deaf children regularly used another’s hand as a tool or some equivalent behaviour, compared with a third of hearing children’s parents reported this behaviour without any combination with any other form of communication. (Table 7).

Parental reporting of the appropriateness of their child’s social responses followed a similar pattern, in that no significant differences were reported in current behaviours. However, when asked to rate children/young people’s most abnormal behaviours when aged four to five, significant differences were noted, with autistic deaf children showing higher rates of significant difficulties in this area (*p* = 0.049). (Table 8).

At group level, no significant differences were recorded in the appropriate integration of eye gaze and communication or social overtures between the two groups. Similarly, no significant differences were reported in the range of facial expressions used by the two groups, in the use of inappropriate facial expressions or in the offering of comfort to others.

### 3.3. Domain B: Communication

#### 3.3.1. Lack of, or Delay in Language Development and Non-Compensation through Gesture

There were no significant differences recorded in children/young people’s use of pointing to express interest, nodding or shaking their head to mean yes and no or using conventional or instrumental gestures either currently or at any time [‘ever’] in their childhood.

#### 3.3.2. Initiating or Sustaining Conversation

Needs-based communication (as indicated by reported levels of social verbalisation/sign/chat) showed no significant differences.

There were no significant differences observed in reciprocal conversation between both deaf and hearing children/young people.

#### 3.3.3. Stereotyped, Repetitive or Idiosyncratic Speech

Items within included under this category related only to children and young people who used verbal or signed communication. Among those children fulfilling these criteria, no significant differences were noted in children/young people’s reported use of stereotyped utterances/signing, either currently or ever. Similarly, no significant difference was noted in reported current use of inappropriate statements or questions. Small, non-significant differences were reported in children and young people’s levels of pronominal reversal.

Parents of children who used sign language were also asked if their children had ever reversed signs. No hearing children participating in the study used sign language. Of the deaf children participating in the study, four were noted to have ever reversed signs (two were currently still doing this), one regularly reversed at least one sign and one was noted to reverse some signs after phrase speech was established. Due to the small number of children exhibiting sign reversal, no statistical analysis was completed on these data.

There were no noted significant differences in children/young people’s use of neologisms or idiosyncratic language, with the majority of participants being rated as 0 or experiencing no to limited difficulties in this area both currently (75% of deaf children with ASD and 58% of hearing children with ASD) and ever (61% of deaf children with ASD and 47% of hearing children with ASD).

Of those autistic children who used sign language, 17% (*n* = 4) were reported by their parents/carers to have sign reversal.

#### 3.3.4. Spontaneous Make Believe/Social Imitative Play

Parents/carers reported that all deaf and hearing children irrespective of their linguistic skills had similar levels in their ability to take part in imitative social play or in their children’s ability to spontaneously imitate the actions of others.

However, when parents were asked to describe their child’s imaginative play when they were aged four to five, autistic deaf children appeared to experience greater difficulties than autistic hearing children (*p* = 0.022) (Table 9).

### 3.4. Domain C: Restrictive, Repetitive and Stereotyped Patterns of Behaviour

Restrictive, Repetitive and Stereotyped Patterns of Behaviour incorporates four sub-domains, these being: preoccupations or circumscribed patterns of interest, non-functional routines or rituals, stereotyped and repetitive motor mannerisms and preoccupations with parts of objects or non-functional elements of material. There were high rates of these behaviours in both deaf and hearing children. However, no significant differences were noted between deaf and hearing participants across any of these domains.

## 4. Discussion

### 4.1. Discussion of Main Findings

The majority of items explored showed no significant differences between autistic deaf and hearing children. For example, there were no significant differences between autistic deaf and hearing children in pre-occupations with abnormal intensity or content, mannerisms/stereotypies, routines/compulsions/rituals and no differences in sensory interests or interests in the non-functional aspect of objects. These appear to be important findings, with these factors being present across both groups in approximately equal amounts. The similarity of these results may indicate that the developmental history is similar for autistic deaf and hearing children. Such similarities may also help to reassure clinicians who have concerns of confidence and competence regarding assessing deaf children, as highlighted by Brenman and colleagues [11]. Of course, it is important to recognise that our study only explored algorithm items and exploration of other items may identify further differences.

Contrary to our hypotheses of greater use of facial expressions in autistic deaf children, neither using facial expressions to communicate or pointing to show things were significantly statistically different between autistic deaf and hearing children. Similarly, though we did see some instances of sign reversal, the number of participants showing this pattern was too small to extract any meaningful statistics.

In line with our hypotheses, there were differences between deaf and hearing children in the peer relationships domain, in that autistic deaf children were less likely to respond to the approaches of other children or play imaginatively with peers. Whilst the former may be, in part, related to not hearing an approach, the latter likely relates to the quality of joint play. Deaf children have been found across a range of studies to face considerable challenges in peer interaction, particularly deaf children in mainstream schools or in environments where the majority of people are hearing [33]. Often, these settings are not able to meet the language and communication needs of deaf children/young people. Autistic children can also face social challenges [34]. The evidence suggests that both autistic children [35] and deaf children more generally [33] seek to make social contacts but autistic children may look for smaller social groups, prefer internet links or making friendships around common interests [36], thus limiting their social contacts. Similarly, deaf children face large communication challenges in hearing contexts [37]. There is no evidence that deaf children generally are any different from hearing children in their desire to play or their engagement in play, but if they are not in a cultural and linguistic environment where play is available to them, that will be a challenge. Further research is needed to explore play in deaf autistic children more carefully. This and other differences between deaf and hearing children may be a further reason why using assessment instruments validated for deaf children, carried out by professionals who can communicate in a child’s first language and who understand the deaf culture (including deaf professionals), should be recommended.

Another area where there was no statistically significant difference related to most aspects of use of language. There were no differences between groups in language delay (with no gesture compensation), conversational reciprocity or unusual or idiosyncratic use of language. As such, our data suggest that the language-related features of autism may be linguistic (i.e., can be signed or spoken) rather than specifically auditory.

Of those autistic children who used sign language, 17% (*n* = 4) were reported by their parents/carers to have sign reversal. As noted above, this study cannot say a great deal about this, given the small numbers, and further research is necessary. Sign reversal has been found in other series [26]. The percentage we found was lower than expected, but this may be because we had a broad range of ages, and sign reversal may only be present within certain developmental linguistic stages, or possibly that some parents/carers were not reporting this finding because they had not noted it themselves (e.g., because of limited sign language skills). This tool is based on parent/caregiver report, thus an adapted play/interaction-based assessment, such as the ADOS-2 Deaf adaptation, is likely to yield more accurate information, especially when the developers have suggested that deaf people fluent in BSL should carry out the ADOS-2 Deaf adaption with deaf participants whose main language is BSL [22].

### 4.2. Strengths and Limitations

One strength of this research is that it is the largest group of deaf and hearing children to date compared internationally. Further research with larger sample sizes needs to take place, although funding for such complex studies in minority groups is not easy to access. There were some limitations in this research. There were very few participating families whose children were aged under four years of age (*n* = 4). However, the ADI-R itself was not designed to study the behavioural profiles of very young children, and so this work needs to be carried out in different contexts. The study was carried out in the UK, where a range of languages are used, singly or together, in various blended ways. Further research in other cultural or linguistic contexts will shine further light on the rich international differences in deaf children and young people.

Further limitations include the fact that the adaptation suitable for deaf children and young people was used across both groups. This was partly because a range of additional questions are asked (e.g., related to complex education and use of different languages) that generated additional information for comparison. Whilst the algorithm is unchanged from the original ADI-R, it is possible that this change is a limitation, but not as serious a limitation as using two different versions for the two different groups, which would have negatively impacted comparability. This was partly for comparative purposes and partly because the items were designed to be able to be used for the full range of deaf participants, including those with mild deafness and those using hearing aids, and so items were chosen to reduce bias against deaf people but to retain the main features of the original ADI-R instrument. We focused on algorithm items in this paper as these are deemed diagnostically important, but this could be a limitation and further research could take a broader view of symptom profiles.

A further limitation of the study is the lack of diversity within our sample. The individuals recruited to the study were predominantly White British and identified as male. This is a persistent issue within autism research, particularly in relation to gender [33,38], and should be taken into consideration when interpreting and making use of these data. Future research may wish to explore the differences between outcomes when using this measure with individuals of varying gender identities.

### 4.3. Implications for Practice

The newly validated ADI-R Deaf adaptation performs well in terms of sensitivity (89%) and specificity (81%), as reported elsewhere [21], and is at least as good as the original ADI-R used in hearing populations. We recommend its use. Some aspects asked about within it were robustly to be found in autistic deaf participants as well as autistic hearing participants, for example, repetitive and stereotyped patterns of behaviour, a common feature of both the World Health Organisation [34,39] and American Medical Association [35,39] diagnostic criteria. Other aspects, such as sign reversal found by other authors [19,26], have been found in our cohort, although may not always be present in signing autistic deaf children. Given that this finding was seen originally in a US sample in children learning ASL, it is important to note this newly reported finding also occurs in BSL (in our sample), which is a different language. This demonstrates that it is not specific to ASL but occurs in more than one sign language. We recommend that practitioners engaged in assessments watch out for this.

Some elements appear to be more common in autistic deaf participants than autistic hearing participants, namely imaginative play with peers. These clinical findings need to be taken into consideration by clinicians in the assessment process.

Another main finding relates to the fact that the experiences of autistic deaf participants appear to align with those of autistic hearing participants and that autism appears to be a helpful construct in the deaf child population. Another important finding is that sign language appears to be equivalent to spoken language in autism presentation.

We would also recommend that there are enough differences between deaf and hearing participants to suggest those assessing deaf children have additional training and include deaf professionals within the assessing team. One model for this is seen in the UK National Deaf CAMHS [25,31].

### 4.4. Implications for Research

Further research is necessary to better understand the presentations of autistic deaf children and young people, including changes in profiles across age ranges and also features less commonly enquired about, such as role shift in BSL. Larger cohorts may allow better differentiation between subgroups of the deaf population, such as those who are profoundly deaf compared to those mild/moderately deaf, and those with different causes of deafness or comorbid diagnoses such as learning disability. Further study across age groups, genders and races may also allow for identification of differences across different stages of development. More specifically, further research in other signed languages could confirm if sign reversal is universal to children with autism in sign languages.

## 5. Conclusions

This study offers the largest comparison of autistic deaf and hearing children and young people to date. Given the focus of the ADI-R on early stages of development, the noted differences that deaf children and young people may experience in their development and the development of an adapted tool, this comparison was able to address multiple research questions. Whilst few significant differences were noted between autistic deaf and hearing children and young people, differences uncovered often related to peer relationships. This paper also offered one of the first reports of sign reversal in autistic BSL using deaf children and young people.

## Figures and Tables

**Table 1 ijerph-20-02143-t001:** Demographics of autistic deaf and hearing children/young people.

		Deaf *n* = 65	Hearing *n* = 41
**Gender**	Male	55 (85%)	28 (68%)
	Female	10 (15%)	13 (32%)
**Age**	0–3	4 (6%)	0 (0%)
	4-9	25 (39%)	20 (49%)
	10+	38 (55%)	21 (51%)
**Ethnicity**	White	49 (75%)	38 (93%)
	Black	1 (2%)	1 (2%)
	Asian	8 (12%)	1 (2%)
	Mixed	6 (9%)	1 (2%)
	Other	1 (2%)	0 (0%)

**Table 2 ijerph-20-02143-t002:** Mean scores for ADI-R Deaf Adaptation Algorithm Domains by Diagnostic Group.

	Deaf ASD		Hearing ASD		*p* Value
	Mean (SD)	*n*	Mean (SD)	*n*	
**A: Reciprocal Social Interaction**	23.7 (4.3)	65	21.7 (6.4)	41	0.079
**B: Communication—Verbal**	16.5 (4.4)	48	17.2 (5.7)	33	0.530
**B: Communication—Nonverbal**	11.9 (2.6)	17	11.8 (3.4)	8	0.877
**C: Restricted, Repetitive and Stereotyped Patterns of** **Behaviour**	6.7 (3.0)	65	7.4 (2.6)	41	0.222

**Table 3 ijerph-20-02143-t003:** Classification based on the cut-off scores as published by Rutter et al. 2003.

		Deaf		Hearing	
		*n*	%	*n*	%
**A: Reciprocal Social Interaction** **(Cut-off = 10)**	No	0	0%	1	2%
	Yes	65	100%	40	98%
**B: Communication—Verbal (Cut-off = 8)**	No	0	0%	3	9%
	Yes	48	100%	30	91%
**B: Communication—Nonverbal** **(Cut-off = 7)**	No	1	6%	1	12%
	Yes	16	94%	7	88%
**C: Restricted, Repetitive** **and Stereotyped Patterns of Behaviour** **Scores in All Three Content Areas Exceed the Specified Cut-Offs, and Onset of the Disorder is Evident by 36 Months of Age.**	No	6	9%	1	2%
	Yes	59	91%	40	98%

**Table 4 ijerph-20-02143-t004:** Differences in direct gaze and social smiling.

		Deaf ASD Current *n* (%)	Hearing ASD Current *n (*%*)*	Deaf ASD Aged 4.0–5.0 *n* (%)	Hearing ASD Aged 4.0–5.0 *n* (%)
**Direct Gaze**	0 = Normal reciprocal direct gaze used to communicate across a range of situations and people	58 (89)	39 (95)	16 (25)	7 (17)
	1 = Definite direct gaze but only of brief duration or not consistent during social interactions	2 (3)	1 (2)	17 (26)	11 (27)
	2 = Uncertain/occasional direct gaze or gaze rarely used during social interactions OR unusual or odd use of gaze	5 (8)	1 (2)	32 (49)	23 (56)
**Social smiling**	0 = Regularly predictable reciprocal social smiles in response to smiles of variety of people besides parent/caregiver	11 (20)	4 (10)	10 (15)	4 (10)
	1 = Some evidence of reciprocal social smiling	20 (31)	19 (46)	10 (15)	9 (22)
	2 = Some evidence of smiling while looking at people and/or smiles only to parent/caregiver, smiles only upon request, smiles in odd situations or odd ways OR little or no smiling at people though may smile at other things	34 (52)	18 (44)	45 (69)	28 (68)

**Table 5 ijerph-20-02143-t005:** Differences in imaginative play with peers.

	Deaf ASD Current (Under 10) *n* (%)	Hearing ASD Current (Under 10) *n* (%)
0 = Imaginative cooperative play with other children in which subject both takes the lead and follows other children in spontaneous pretend play	0 (0)	1 (6)
1 = Some participation in pretend play with another child but not truly reciprocal and/or pretending is limited in variety	5 (19)	10 (59)
2 = Some play with other children but little or no pretending OR no play with other children or no pretend play even on own	21 (81)	6 (35)

**Table 6 ijerph-20-02143-t006:** Response to the approach of other children.

	Deaf ASD Current (Under 10) *n* (%)	Hearing ASD Current (Under 10) *n* (%)	Deaf ASD Aged 4.0–5.0 *n* (%)	Hearing ASD Aged 4.0–5.0 *n* (%)
0 = Generally responsive to other children’s approaches although may be hesitant initially if other children are too rough/intrusive. Makes a clear effort to keep an interaction going with a child other than a sibling by gesturing, vocalising, signing, offering an object.	3 (12)	9 (53)	4 (6)	7 (18)
1 = Sometimes responsive to other children’s approaches but response is limited, somewhat unpredictable or only to a sibling or a very familiar child.	12 (48)	7 (41)	20 (32)	19 (48)
2 = Rarely or never responds to the approach of even a familiar child OR consistently and persistently avoids approaches of other children	10 (40)	1 (6)	38 (61)	14 (35)

**Table 7 ijerph-20-02143-t007:** Use of another person’s body to communicate—Ever.

	Deaf ASD *n* (%)	Hearing ASD *n* (%)
0 = No use of other’s body to communicate expect where other strategies have not worked or when taking someone’s hand to lead them places	25 (39)	14 (37)
1 = Occasional placement of other’s hand on objects or use of other’s hand as a tool or to point but some combination with other modes of communication	6 (9)	12 (32)
2 = Occasional placement of other’s hand or use of other’s hand as tool or to demonstrate for subject without integration with other modes of communication OR regular use of other’s hand as a tool or to gesture for the subject	33 (52)	12 (32)

**Table 8 ijerph-20-02143-t008:** Appropriateness of social responses.

	Deaf ASD Current *n* (%)	Hearing ASD Current *n* (%)	Deaf ASD Most Abnormal Aged 4.0–5.0 *n* (%)	Hearing ASD Most Abnormal Aged 4.0–5.0 *n* (%)
0 = Appropriate response to overtures by familiar and unfamiliar adults	3 (5)	2 (5)	0 (0)	2 (5)
1 = Some clear positive responses and interactions but not consistent	33 (51)	24 (60)	11 (18)	12 (31)
2 = Responds to parents/caregivers and others in familiar settings but responses are stereotyped, inappropriate or very limited OR little or no interest in or response to people except parents/caregivers or very familiar significant others	29 (45)	14 (35)	51 (82)	25 (64)

**Table 9 ijerph-20-02143-t009:** Differences in imaginative play.

	Deaf ASD Aged 4.0–5.0 *n* (%)	Hearing ASD Aged 4.0–5.0 *n* (%)
0 = Variety of pretend play including use of dolls/animals/toys as self-initiating agents	0 (0)	4 (10)
1 = Some pretend play including actions direct to dolls or cars but limited in variety or frequency	10 (16)	3 (8)
2 = Occasional spontaneous pretend actions or highly repetitive pretend play (which may be frequent) or only play that has been taught OR no pretend play	52 (84)	33 (83)

## Data Availability

No new data were created or analysed in this study. Data sharing is not applicable to this article.

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
