# Peer review of "Comparison of Diagnostic Profiles of Deaf and Hearing Children with a Diagnosis of Autism"

_ijerph, 2023, doi:10.3390/ijerph20032143_

Round 1
Reviewer 1 Report
Thank you for allowing me to review the manuscript. The paper highlights the need for increased awareness and research in a population that has thus far received little attention or services. In my review, I sequentially reviewed the manuscript, noting both grammatical issues and revision recommendations. To illuminate the more important areas/significant concern, I highlighted those in RED.
Abstract
Line 17, remove number 6
Introduction:
Line 27: Spell out 9 in 9%
Line 30: delete “are autistic, also referred to in practice and literature as”, change “having” to have.
Line 31: Delete “The diagnosis of”, start sentence with Autism.
Line 34: Need transition sentence between sentence ending on line 34 and new sentence beginning. You jump from co-morbidities to there is limited research, need a transition sentence.
Line 36, start of sentence: Presentation of what? Autistic behaviors? Need to specify what “comparisons of presentation” s referring to.
Line 36: Need to address the importance of presentation of behaviors among hearing/deaf when assessing for autism. How does a specialized assessment help with diagnosing? I am looking for a rationale on why a specific assessment tool is needed rather than taking administering the gold standard assessment in sign language.
Line 38: You mention that delay in diagnosis occurs in this population. First, please explain why you labeled deaf autistics as an at-risk group. What puts them at a higher risk? Second, please explain the “regularly occur” statement. Do you have comparison data when considering diagnosis of children in the UK?
Line 38: You compare data from US and UK, but your focus is on the UK. Hence, I recommend stating why you are using comparative data from the US when in the previous sentence you are discussing autistic children who are deaf. My recommendation, move the US data up earlier
Line 41: What is “general population” referring to, US and UK or just UK? My recommendation—move the US and UK comparative data sentence up closer to beginning of paragraph, then discuss just the UK and diagnosis discrepancies (about line 40-42), then discuss differences in assessment between hearing and deaf autistics.
Line 46-47: Combine the citations (13, 14, 15) at end of sentence, helps with sentence flow.
Line 50: It is imperative you explain the barriers, at the very least list out the barriers. Barriers to assessment/diagnosis is an important piece of your paper, since you mentioned lack of assessment/diagnosis within the deaf community. Also, explaining if these are similar or different barriers than hearing autistics encounter.
Line 55: Delete “have needed to” and change “rely” to relied. This is a passive sentence, make it active by stating “most relied…”
Materials and Method:
Line 76: Replace “to” with “by”. Participants are recruited by an agency or a group, they are not recruited to a group; that would be referred. Recruit means to bring in.
Line 92: Space needed between Chicken and parenthesis. I noticed numerous spacing issues (needs a space between words), please read through entire manuscript and revise the spacing issues.
Line 98: Change “See Below” to more specific wording, such as “See Section ABC” or “See Table 1”. On my copy, the “See Below” is at the end of page 2. Hence, directing your reader to a specific area of your paper is clearer directions.
Inclusion and exclusion:
Line 100: Delete “Participating children/young people” to “Participants”
Line 101-102: Please replace “be deaf with at least 40dBHL” to something like “possess hearing range no higher than 40dBHL.” After you explain who the participants are (children/young people), use term “participants” in remainder of manuscript.
Instrument:
Line 110-111: I am using these sentences as examples; in manuscripts always use past tense not current. The study was already completed; hence we use past tense. “The interview is conducted…” is replaced with “The interview was conducted…”
Line 111: Move “usually a parent/caregiver” on line 112 to line 111 after “with an informant”. It would read, “with an informant, usually a parent/caregiver, who knows the participant very well.”
Procedural Analysis:
Line 126: You state children without a diagnosis of autism were recruited. However, in your Exclusion criteria you mention participants were required to have evidence of an autism diagnosis. This is a major flaw that needs to be explained.
Line 126: Replace “main instrument” with actual name of the instrument.
Lines 126-128: Move these to Line 108, this is about the instrument.
Line 130: Add acronym (ADI-R Deaf Adaptation) in Line 108. Allows reader to know what the acronym is referring to.
Results:
Table 3, Lines 171-177, and Table 4: Explain what “most abnormal” refers to. Providing an example of “most abnormal” would be helpful.
Lines 265-270: Need to add data, since (a) you stated that no significant differences were found—that means you collected and analyzed data; (b) you included data on this domain in your discussion and (c) restrictive and repetitive behaviors is a cornerstone behavior of autism, hence if majority of your participants were not engaged in this behavior my main question would be “Why not?”
Discussion:
Line 291-305: Need to expound on why autistics who are deaf not engage in peer relationships or play the same as autistics who can hear. Need to explain how deafness would not be a factor in the differences. Do deaf children play differently than hearing children? If that is the case, then autism itself may not be causing the play differences, it is being deaf. That would be a good rationale supporting the need for a specific assessment tailored for non-hearing children.
Line 311-312: Need to put this data in results (I mentioned this previously in my review). Also need to explain the importance of this, why is focusing on sign reversal important to study? Many of your readers may not know what sign reversal is—need to give a definition—followed by the importance of noticing sign reversal. Since you mention this topic at least three times (results, discussion, and future research), I would say sign reversal deserves more than just a glancing review.
Lines 334-346: At the very start of your manuscript, you state a purpose of this paper was to compare diagnostic assessment symptoms (lines 57-61). Additionally, you state in Lines 126-128 a purpose is to analyze the sensitivity and specificity of the instrument. In your discussion section, you do not specifically address either of these. I did not see a summary explaining how well you thought the assessment did in assessment. Discussing the data first is great; please follow up with your own discussion (based on the data) how well you thought the assessment, and areas of improvement.
Reviewer 2 Report
The text is interesting and the report on the study is useful, but for a narrow range of experts.
Recommendation for authors:
- Explain the abbreviation CYP in the abstract as it is commonly used
- Use deaf and hard of hearing children (DHH) rather than just deaf. Also, the term deaf is used inconsistently: deaf and Deaf, which fundamentally changes the terminological concept
- More recently, the term children with typical hearing have been required instead of hearing children (this is more a matter of editorial perspective, I don't agree with this wordplay)
Round 2
Reviewer 1 Report
Thank you very much for the work and time you put in with the revisions.